# Polytypes and planar defects revealed in the purine base xanthine using multi-dimensional electron diffraction
Helen W. Leung [1] ✉, Royston C. B. Copley [2], Giulio I. Lampronti [1], Sarah J. Day[3], Lucy K. Saunders [3], Duncan N. Johnstone[4] & Paul A. Midgley [1] ✉

Layered crystal structures are commonly found across organic and inorganic material systems. When in-plane atomic arrangement remains (nearly) identical, a stacking variation of these layers may result in twinning, planar disorder, or polytypes, a form of polymorphism derived from altering stacking sequences. In this work, we use multi-dimensional electron diffraction (ED) modalities to explore the microstructure of xanthine, an archetypal purine base with a layered crystal structure. Firstly, we identify and characterise the twin operator relating domains of Form I xanthine. We then solve the structure of a new xanthine polymorph, revealing that it is a polytype of Form I. Finally, interfaces between twin and polytype domains are visualised, whilst streaking in the diffraction patterns reveals the presence of planar disorder. Given these observations in the xanthine system, this work suggests that disorder on the nanoscale may be a commonly occurring phenomenon in layered organic molecular crystals.

Layered crystals are a wide class of materials that span inorganic and organic systems of high scientific interest owing to their unique functional and physicochemical properties[1]. Crystals with layered structures lend themselves to the possibility of variations in the stacking sequence of their layers whilst intra-layer atomic coordination remains fixed; this can manifest as (1) twinning, (2) polytypism, or (3) planar disorder.

Firstly, twinning refers to the intergrowth of two or more differently oriented crystal domains that are related by a common plane or direction, and is an important feature in the microstructure of a material. Twinning is of interest in a number of biogenic crystals, such as the purine base guanine, for its use as a mechanism for biological control of optical properties[2,3]. Secondly, polytypism is a form of polymorphism found in layered structures in which low-energy alternate stacking configurations can be adopted by the translation of layers with no change to the intra-layer atomic coordination[4,5]. In inorganic systems, different polytypes may display different functional properties, such as in (bilayer/multi-layer) graphene[6,7], or transition metal dichalcogenides[5]. Thirdly, planar disorder refers to translational displacements between consecutive layers in the crystal, which may, in principle, encompass a range of displacements, from well-defined to random. As displacements between layers become less random (a scenario defined as 'order-disorder' polytypism by Dornberger-Schiff[8,9]), geometric equivalence on a local scale is restored, but the crystal's long-range symmetry is still disrupted[10]. Stacking faults are a type of planar defect in which one atomic

plane is stacked out of sequence with another via a defined vector, disrupting the continuity of an otherwise perfect lattice. Understanding the interactions between the three phenomena of twinning, polytypism, and planar disorder builds an important understanding of the microstructure in a layered material. Exploration of these concepts in organic molecular crystals is still an emerging field, but these insights may play a key role in mechanisms of crystal growth and resulting mechanical and physicochemical properties in organic crystals[11–14].

Xanthine (3,7-dihydropurine-2,6-dione) belongs to the purine family and is made of five- and six-membered heterocyclic rings: this chemical skeleton is also found in other purines, such as guanine. It serves as an intermediate in nucleic acid metabolism, acting as a precursor for the synthesis of uric acid, and is therefore widely found in organisms and of interest in a pharmacological context[15]. In previous work[16,17], 3D-ED was used to elucidate the structure of xanthine (Form I). The Form I xanthine structure consists of hydrogen-bonded layers with weak interlayer van der Waals' forces (Fig. 1). However, X-ray powder diffraction (XRPD) of the bulk sample shows asymmetric peak profiles and the presence of peaks which do not match the Form I unit cell, indicating the presence of both planar disorder and other phases, respectively[16]. Stacking disorder has also previously been suggested as the reason for differences between experimental and simulated XRPD profiles in layered organic pigments[18,19]. As a fundamental organic molecule with a layered structure, xanthine serves as

[1]Department of Materials Science and Metallurgy, University of Cambridge, Cambridge, United Kingdom. [2]GSK R&D, Stevenage, United Kingdom. [3]Diamond Light Source Ltd, Beamline I11, Harwell Campus, Didcot, United Kingdom. [4]GSK R&D, Upper Providence, USA. ✉e-mail: hl585@cam.ac.uk; pam33@cam.ac.uk

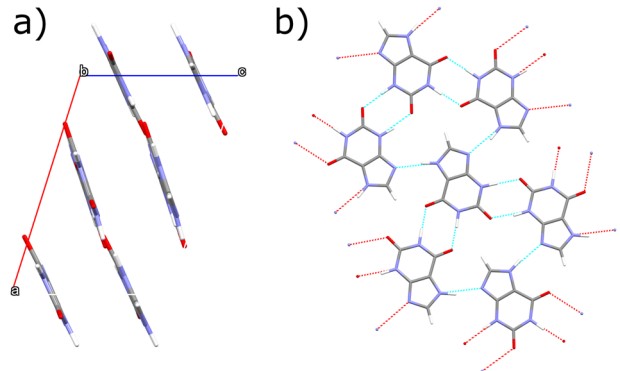

**Fig. 1 | Xanthine Form I structure. a** [010] projection, showing stacked layers of xanthine molecules. **b** The intra-planar hydrogen-bonded network of xanthine molecules[16].

an exemplar system to study the structural behaviour of this class of layered organic molecular crystals.

Transmission electron microscopy (TEM) has been used for decades to establish a fundamental understanding of crystal structure and microstructure across a broad range of (inorganic) materials. However, the beam-radiation sensitivity of organic crystals has, until recently, limited the application of conventional TEM techniques to explore the microstructure of these systems. As a result, the characterisation of microstructural features such as defects in organic crystals is a relatively nascent field[20,21]. Different electron diffraction acquisition modalities can overcome challenges associated with structural characterisation of organic crystals, namely: (1) 3D electron diffraction (3D-ED)[22,23], and (2) 4D scanning transmission electron microscopy (4D-STEM)[24]. Over the past decade, 3D-ED has developed as a powerful technique for structure determination of micron-sized organic crystals[22,23], and is of particular importance in instances where material is scarce, or single crystals of sufficient size for single-crystal X-ray diffraction are difficult to grow. In 3D-ED, parallel illumination, with a beam typically ca. 1 micron in diameter, is used to collect diffraction patterns. For heterogeneous regions of interest with perhaps varying domain structures, the resultant pattern will be a superposition of the diffracted signal from across those domains. By contrast, 4D-STEM[24] uses a quasi-parallel (diffraction-limited) beam a few nm in diameter to raster scan across a region of interest. A 2D diffraction pattern is acquired at every pixel in the 2D raster scan, thus creating a crystallographically-rich 4D dataset. Once acquired, diffraction patterns from sub-regions can be determined, orientational variations in those patterns plotted across regions of interest, and virtual bright-field or virtual dark-field (VDF) images reconstructed by plotting the spatial variation of a particular diffracted intensity (chosen using a virtual objective aperture).

In this work, we use electron diffraction to explore polytypism and the defect microstructure in xanthine crystals. Firstly, twinning about the [101] axis is observed in Form I. Secondly, we determine a new phase of xanthine (Form II). Using 3D-ED to elucidate its structure, we show that Form II is a polytype of Form I. Next, we apply 4D-STEM to highlight the spatial relationship and interfaces of twin boundaries and intergrowths between polytypes. Here, VDF imaging and distinct streaking in diffraction patterns indicate the presence of rigid-body translational planar disorder within domains. These insights are supported via a multiphase Rietveld refinement (incorporating a stacking fault model) of in situ high-resolution synchrotron XRPD data to demonstrate consistency between electron diffraction data and that from the XRPD of the bulk sample.

## Results
### Twinning in xanthine
Samples from commercially acquired xanthine, synthetically produced by deamination of guanine[25], were deposited onto TEM grids (see S1). 3D-ED datasets from multiple sub-micron-sized crystals of xanthine were collected

and processed, with each comprising a tilt series of diffraction patterns (as described in Methods). In some datasets from a single particle (Fig. 2a), the presence of two separate reciprocal space lattices was observed upon the reconstruction of the 3D reciprocal space (this is further described in Supplementary Note 2). The low overlap of reflections enabled successful integration and structure solution, treating the reflections from each reciprocal lattice as independent, thus splitting the dataset into two. Each dataset was consistent with Form I xanthine[16].

Twin domains are related by a symmetry operator[26], defined as the twin law. To preserve the strong intra-planar hydrogen bonding, any twinning mechanisms in xanthine likely involve interlayer displacement, which disrupts only the weaker van der Waals' bonding[27,28]. To determine whether the xanthine lattices were indeed twinned (as opposed to just two misoriented crystals), the orientation matrices were compared. This revealed an orientation relationship which can be described using either a rotation of 180° or a reflection about (10$\bar{1}$), confirming a twinning mechanism and a twin law in xanthine. It is not possible to distinguish between the rotation and reflection operations in projection, given that Form I xanthine (with $P2_1/c$ space group) has a centro-symmetric structure. This analysis and the notation to describe this twin law are given in greater depth in Supplementary Note 3. The action of the twin preserves intra-layer molecular arrangements between twin domains (Fig. 2c) and acts to flip the layers. Since the composition surface (the surface along which the lattice points between domains are shared) is parallel to each other, this is a polysynthetic twin and may lend itself to multiple twinning[29,30].

### A new polytype of xanthine, Form II
Most xanthine particles picked for 3D-ED investigation were found to have unit cells consistent with Form I, regardless of whether they were found as single or twinned crystals. However, in one particle of xanthine (shown in the inset in Fig. 3d), two different crystal domains were seen: one of the domains had lattice parameters consistent with Form I xanthine whilst the second domain had orthorhombic cell parameters: [$a = 10.10(10)$ Å, $b = 12.54(10)$ Å, $c = 17.91(17)$ Å, $\alpha = 90°$, $\beta = 90°$, $\gamma = 90°$, $V = 2269(36)$ Å$^3$]. Reciprocal space sections of the orthorhombic lattice (Fig. 3) were used to observe systematic absences of reflections and deduce possible space groups (further described in S4).

Integration was carried out using the 'dual crystal' mode in CrysAlisPro, which produced a set of intensities accounting for the overlapped reflections from the Form I phase (only 11 out of 455 reflections corresponding to the new orthorhombic form were reported to overlap). A structure solution for the orthorhombic phase was successfully achieved using ab initio Dual Space methods implemented in SHELXT (with 500 trials, specifying the $P2_12_12_1$ space group), $Z' = 4$, $Z = 16$, with a 79% completeness. All non-hydrogen atoms were found in the initial structure solution. A kinematical refinement using the least-squares matrix in SHELXL resulted in a final structure, which we call Form II, with an R-factor of 17% to 0.9 Å resolution (corresponding to $R_{sigma} = 0.47$). Further information regarding structure solution and refinement of Form II xanthine is included in Supplementary Note 4.

Form II is closely related to Form I but has higher point group symmetry. Observed reflections that are common to both forms are highlighted in blue in Fig. 3b), with the remaining extra reflections observed resulting from the higher symmetry Form II. A comparison of Form I and II is shown in Fig. 4. Molecular arrangements within hydrogen-bonded layers are identical between forms (Fig. 4b, d), with these layers in xanthine bearing a strong resemblance to the bonding found in hypoxanthine[15,31]. In both forms, hydrogen-bonded layers are stacked. The direction of layer stacking is described by the normal to the (10$\bar{1}$) planes in Form I and by the normal to the (010) planes in Form II. The difference between Form I and II xanthine lies in their interlayer stacking sequence. Therefore, Form II can be described as a polytype of Form I. Different polytypes may arise when multiple stacking arrangements lie within a shallow energy landscape, and there are multiple energetically favourable configurations[32]: in xanthine, the

**Fig. 2 | Twinning in Form I xanthine. a** TEM image of the twinned crystal from which 3D-ED data were collected. **b, c** Illustration of the proposed Form I xanthine interlayer twinning mechanism. Here we observe a 180° orientation relationship between the two lattices, which can be described as a rotation twin axis ~ [1 0.1 1]. This rotation axis lies in the hydrogen-bonded (10$\bar{1}$) planes. Such a twin axis ensures the normal to the (10$\bar{1}$) planes remains invariant under the action of the twin; therefore, the (10$\bar{1}$) planes remain parallel (or anti-parallel) when going from parent to twin.

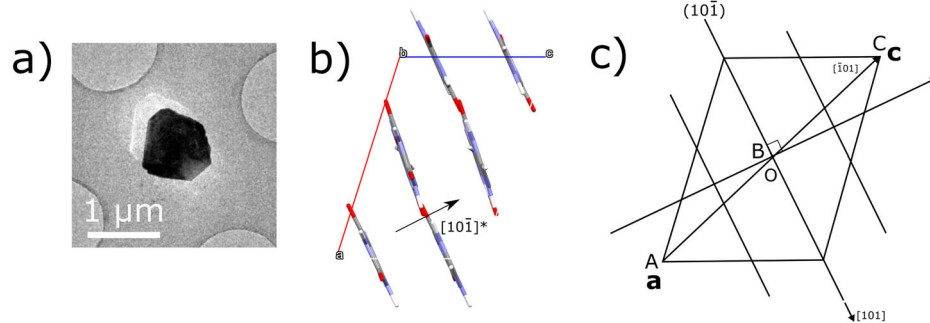

**Fig. 3 | Reciprocal space sections of Form II xanthine from 3D-ED. a** *hk*0, **c** 0*kl*, **d** *h*0*l* sections for Form II xanthine from reconstructed reciprocal space. Systematic absences are consistent with the $P2_12_12_1$ space group. **b** The *h*0*l* section from monoclinic Form I (from a separate crystal) is overlaid onto the orthorhombic Form II *hk*0 section to compare their similarities. The larger unit cell in Form II results in an increase in the number of reflections, with a limited number of reflections overlapping with those of Form I. The inset in **d** shows the micron-sized bi-crystal of xanthine from which this dataset was obtained. Sections shown in this figure come from the Form II domain of the crystal. Reflections from the Form I domain are largely not seen in these sections, except in the far left of the *h*0*l* section. The jagged nature of the left straight facet of the particle hints at its bi-crystalline nature.

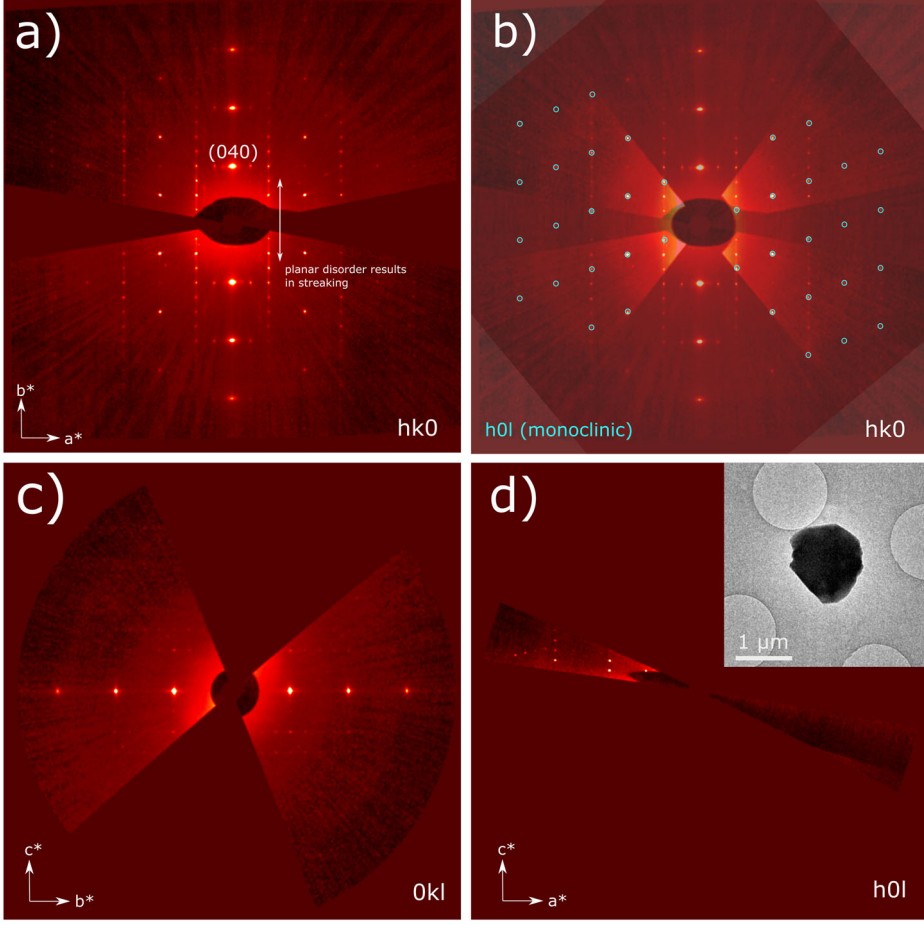

case of a shallow energy landscape is supported by density functional theory (DFT) modelling approaches for variations on its layered structure[33].

We inspect Form I and II by mapping the movement of xanthine molecules between consecutive layers in both structures. Interlayer translations in both forms can be simplified into two preferred relative positions of the xanthine molecule, shown in Fig. 5i) by vectors **A** and **C** (= -**A**), or **B** and **D** (= -**B**). **A** and **B** are also related by mirror symmetry (Fig. 5iii). Vectors **C** and **D** are defined to minimise the magnitude of the vectors (further details on how these vectors are defined are shown in the Supplementary Note 4, Fig. S4.2). Movement between adjacent layers may be described by one of these four vectors. As shown in Fig. 5ii) and iv), in Form I, the layer stacking follows repeated application of vectors **A** and **B**. Layer positions do not repeat after any set number of layers, consistent with the

monoclinic symmetry of the cell. In Form II, the stacking of these layers follows the sequence **A B C D**, with atom co-ordinates repeating every four layers.

Different combinations of the four possible observed stacking vectors (**A**, **B**, **C**, and **D**) will lead to varying degrees of short to long-range order in xanthine crystals. This behaviour is consistent with observed streaking in reciprocal space (Fig. 3a). The observation of streaking in diffraction points to the presence of planar disorder. To further explore this, 4D-STEM was applied to the crystals.

**Varying degrees of planar disorder**

Whilst orientation relationships between twin or polytype domains can be elucidated using 3D-ED, information solely in reciprocal space is limited in

**Fig. 4 | Structures of Form I and Form II xanthine.** **a** [010] projection of Form I xanthine. **b** Hydrogen-bonded layers in Form I xanthine, which are described by (10$\bar{1}$) layers. **c** [001] projection of Form II xanthine. **d** Hydrogen-bonded layers, in Form II xanthine, which are described by the (001) planes, have identical coordination to the layers found in Form I xanthine, although the stacking sequences between layers shown in (**a**) and (**c**) are different.

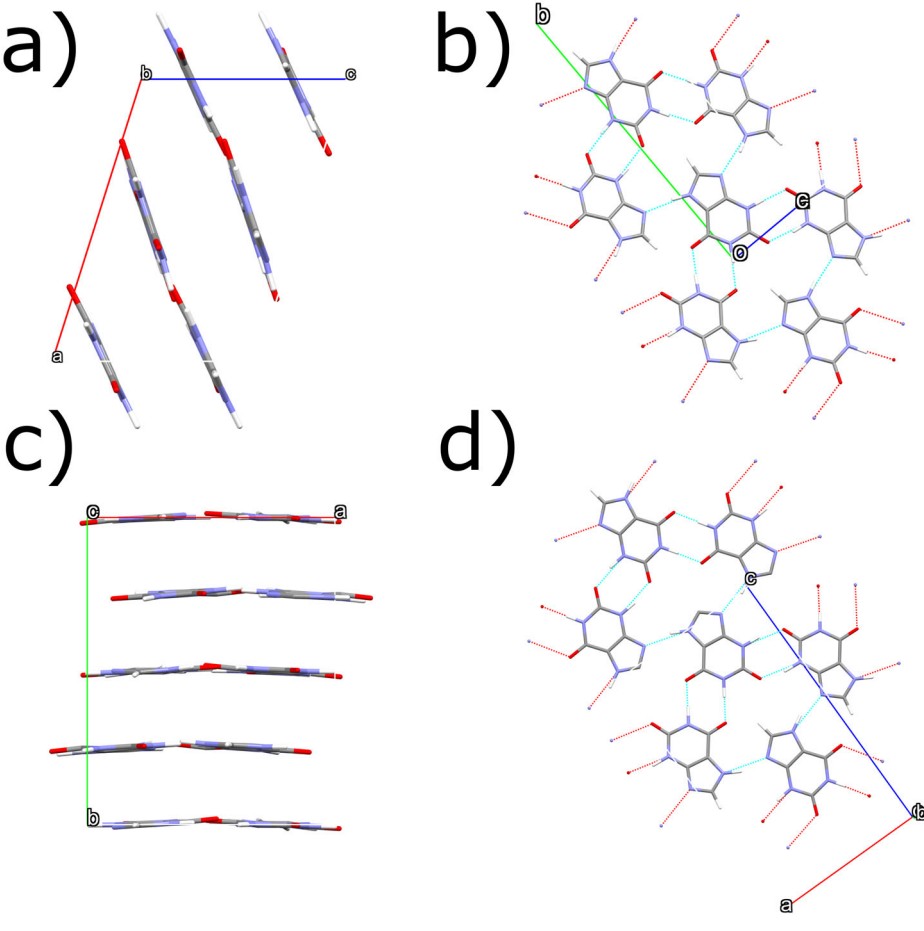

its ability to describe the spatial distribution, the number of these domains (although calculating twin fraction is possible and well-established[34]), or the nature of associated interfaces. Here, 4D-STEM is used to visualise the xanthine microstructure directly. Particles typically appeared blocky in morphology, often with a pair of parallel facets and a pair of less well-defined edges (such as in Fig. 6b). The resulting 4D dataset is analysed in two ways: (1) summed diffraction patterns from sub-regions of the crystal are used to understand structural variations and orientation changes; (2) VDFs are formed using the intensity from a specific Bragg reflection mapped across the region of interest; this highlights in particular which areas are at the Bragg condition for that particular reflection and thus can highlight twins and planar disorder.

Firstly, diffraction patterns were summed over small local areas of the xanthine particle, chosen by manual inspection of the datasets. This improved the signal–to-noise from a given area and highlighted any streaking (e.g., Fig. 6c), akin to that seen in 3D-ED data (e.g., Fig. 3a). The streaking observed is consistent with that previously seen in guanine[35], layered organic pigments[18,19], and other systems[36–40]. The streaking in reciprocal space is parallel to the direction defined by the normal to the stacked layers (as depicted in Fig. 6a). However, there is no streaking through the reflections in the strong systematic row that passes through the origin, i.e., the (h0-h) reflections in Form I and (0k0) reflections in Form II. This is consistent with a rigid-body transverse displacement of the layer with respect to the expected sequence, i.e., very much akin to a stacking fault, whose stacking fault vector is likely to be equal to **A B C** or **D**. The continuous streaking seen implies that these 'stacking faults' occur at near-random intervals.

In some material systems, this streaking may be so severe that a complete set of reflection intensities cannot accurately be measured[41]. In our case, however, the sharpness of individual reflections suggests we have a relatively large fraction of ordered phase with a modest number of stacking faults.

Figure 6c–f are summed diffraction patterns from isolated areas of the particle, highlighted with corresponding colours in Fig. 6b. Indexing these patterns was not straightforward for two reasons. Firstly, the low symmetry of the xanthine unit cell leads to many possible orientations with similar pattern geometries. Relying on simulated intensities to distinguish between orientations based on structure factors is also unreliable, given the sensitivity of diffraction data to any changes in orientation, exacerbated by small Bragg angles in the crystal.

Nonetheless, where possible, indexing patterns revealed the complex microstructure of xanthine particles as hinted by 3D-ED data. Ambiguity in indexing was minimised by the identification of the strong, non-streaked reflections corresponding to the layer stacking, fixing one vector of the diffraction pattern. As a result, adjacent domains highlighted in Fig. 6g, h can be indexed (Fig. 6c, d) as Form II and Form I polytypes of xanthine, separated by a distinct flat interface parallel to the molecular layers. This shows that different polytypes may be adjacent in one particle, in addition to any twin domains. This observation of multiple phases / domains within a particle is consistent with 3D-ED data in which multiple lattices were observed in reciprocal space. Reciprocal lattice vectors corresponding to [10$\bar{1}$]* in Form I and [010]* in Form II are consistently oriented such that they are normal to the flat habit plane (surface) of each particle. The variable extent of streaking across different summed patterns demonstrates that planar disorder is not homogeneous across all domains of the particle. By using virtual apertures centred on certain reflections, the VDFs in Fig. 6 h–k show clear diffraction contrast that corresponds to distinct phases or domains at different positions within each particle. The interface of each highlighted domain is always parallel to a distinct crystal edge and parallel to the layers. Within each domain, fine-scale 'stripy' VDF contrast suggests the

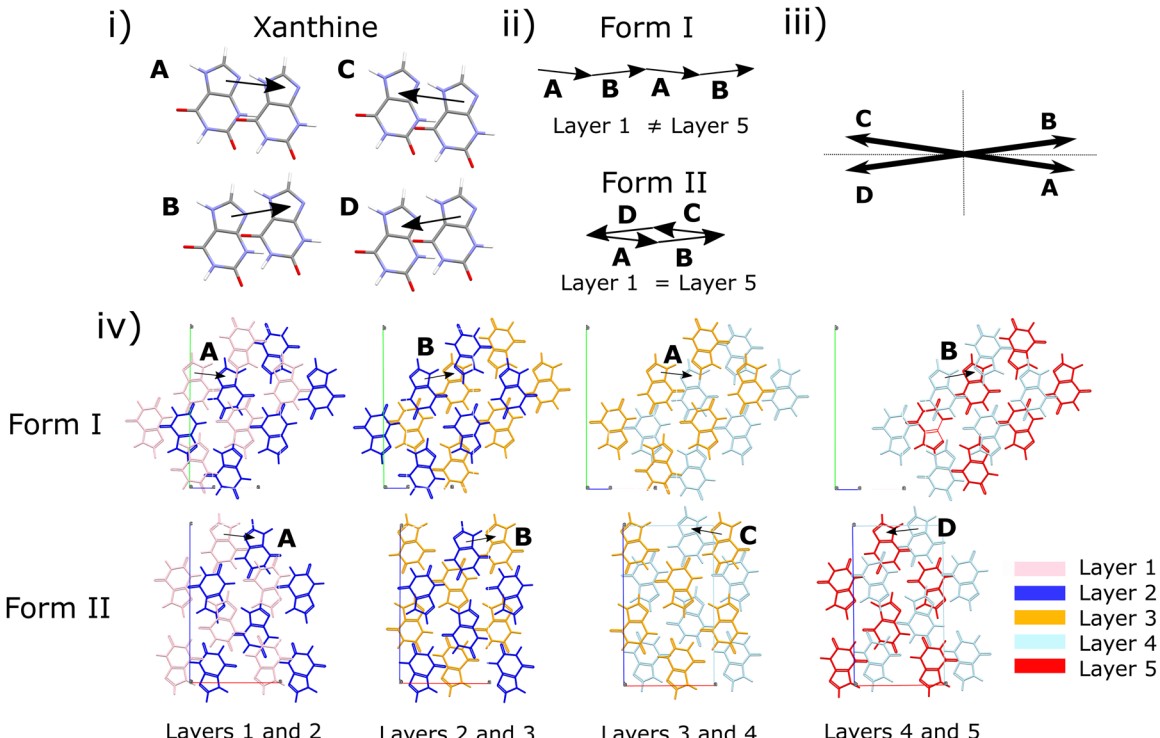

**Fig. 5 | Interlayer translations within the xanthine crystal structure. i, ii** The translations between layers in xanthine can be described by: **A B A B** (Form I), **A B C D** (Form II). **iii a** and **c** components of the vectors **A B C** and **D** are shown: these vectors share the same magnitude (0.377**a**, 0.250**b**, 0.041**c** in terms of the Form II cell) but the direction of **a** and **c** components can be positive or negative, with the vectors related by mirrors as depicted by dotted lines. **iv** layer-by-layer translations highlighted across 5 consecutive layers in Form I and Form II. Layer 1 (pink), Layer 2 (dark blue), Layer 3 (orange), Layer 4 (light blue), Layer 5 (red) = Layer 1 (pink) in Form II. In Form I, which has lower symmetry, Layer 5 (red) ≠ Layer 1 (pink).

presence of local planar disorder, consistent with streaking observed in diffraction space. Another further example of a particle of xanthine with similar contrast is described in Supplementary Note 5.

### Characterisation of bulk material using XRPD

XRPD was used to establish consistency between the individual particles studied with electron diffraction and the bulk powder sample. In-situ XRPD data previously collected from the Diamond Light Source Synchrotron Beamline I11[42] were re-analysed, now including fresh insights from our electron diffraction experiments to account for the new polytype (Form II) and stacking disorder. The presence of multiple phases and stacking faults within individual nanoparticles explains the difficulty in achieving a successful structure solution solely through XRPD methods. This is compounded by difficulties in distinguishing between forms due to their structural similarity, resulting in shared peaks.

Anisotropic line broadening asymmetry is evident in the XRPD pattern. This is a sign of the presence of planar defects, which causes profiles of Bragg reflections to broaden and shift differently depending on $hkl$ selection rules[43–45]. This pattern characteristic has been commonly observed in minerals such as expandable clay minerals, graphitic carbon, boron nitride, and $MoS_2$[46,47]. Peak profiles affected by planar faults may also be convoluted by other line broadening effects[48] such as small crystallite size[49], in the stacking direction in particular.

Firstly, multiphase Rietveld refinement of XRPD data was carried out, including both Form I and Form II xanthine structures. This led to an improved fit, giving an $R_{wp}$ of 2.36%, and Goodness of Fit (GooF) of 3.72%, with all peaks accounted for (Supplementary Note 6, Fig. S6.1).

To further improve our XRPD model, we created an enlarged unit cell (a supercell) to simulate faulted xanthine structures[50–52]. Our protocol (described in Supplementary Note 6) allows for the random introduction of stacking faults to a Form II supercell made of 20 layers (5 unit cells), creating

a model of planar disorder. This inclusion of stacking disorder improved the model, in particular in fitting to the asymmetric peak profiles. This model was refined to an $R_{wp}$ of 1.55% and GooF of 2.46% (Fig. 7).

### Discussion

The nature of different degrees of planar disorder in xanthine stems from stacking variation between its hydrogen-bonded layers. As a small organic molecule with an archetypal purine ring structure, insights derived here may also be applicable to a range of other molecular crystal systems. Cytosine[53], thymine[54], guanine, adenine, uracil[55], eniluracil[56], caffeine, and theophylline are key examples amongst a large class of inter-related molecules which have layered structures that may exhibit similar behaviour. We discuss and compare interlayer stacking in some of these systems.

In xanthine, a shallow energy landscape likely leads to the variety of structural behaviour of the hydrogen-bonded layers. Applying different translations between adjacent layers leads to the Form I (**A B A B** translation) and II (**A B C D** translation) polytypes (Fig. 8i), whereas applying a rotation about [101]/reflection about (10$\bar{1}$) in Form I flips the layers, leading to twinning. To consider the similarities between polytypes and twins in xanthine, we studied the stacking sequence in the known crystalline (triclinic) form of hypoxanthine, another product of the purine degradation pathway. Hypoxanthine has an intra-layer structure highly similar to that of xanthine, deriving from similar hydrogen-bonding interactions[15,31,42]. However, hypoxanthine has different interplanar stacking (Fig 8ii). In its triclinic form, the relationship between adjacent layers is described by a translation combined with a 180° rotation, which alternates between each layer about two perpendicular axes (one axis coming out of the page in Fig. 8ii **A**, one axis in the plane of the page in Fig. 8ii **B**). This stacking sequence is reminiscent of the twinning mechanism observed in xanthine; both offer a way to alter the stacking of hydrogen-bonded layers. The preferred tendency for layers of hypoxanthine to flip suggests a lower associated

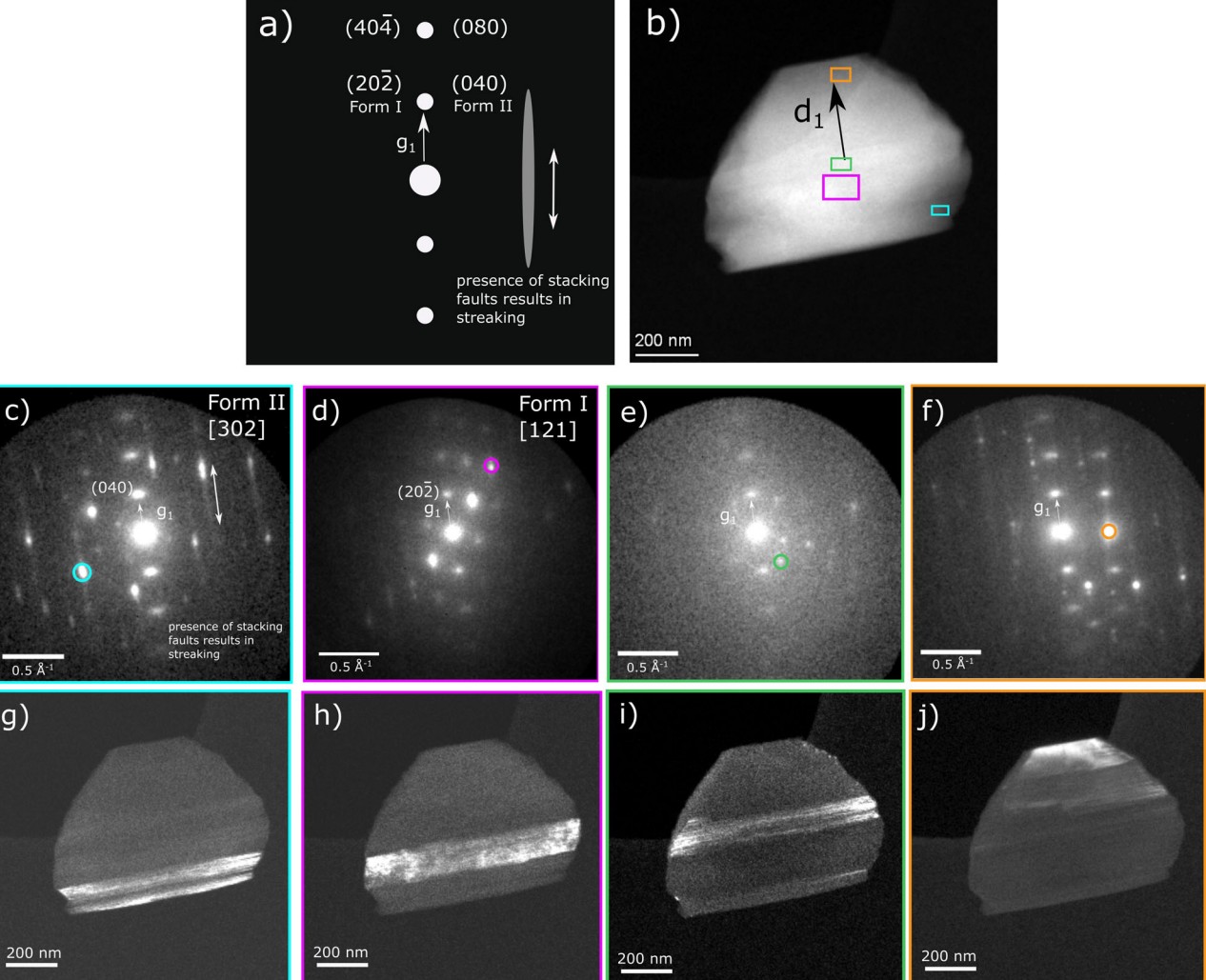

**Fig. 6 | 4D-STEM of xanthine. a** A schematic to highlight the streaking in diffraction, most visible in (**c, f**). The interlayer distance corresponding to $g_1$ stays constant even as layers are laterally displaced. **b** High Angle Annular Dark Field (HAADF) image of a xanthine crystal. Diffraction signals from the areas highlighted in different colours are used to form the summed diffraction patterns shown in (**c–f**). Virtual Dark Field (VDF) images **g–j** are formed from apertures circled in these summed patterns and are highlighted in corresponding colours. VDF contrast indicates the presence of domains within the nano-crystals with interfaces parallel to the stacked layers.

energy cost than in twinning in the xanthine structure. Interestingly, selected area electron diffraction of hypoxanthine particles at different stages of maturation also exhibited a range of similar patterns[57]. These patterns range from: (1) only showing diffraction spots relating to the interlayer stacking (akin to Fig. S5.1c); (2) diffraction patterns with streaking parallel to the stacking direction (akin to Figs. 6c and S5.1d); and (3) indexable, non-streaked electron diffraction patterns (akin to Fig. 6d and S5.1b). In the case of hypoxanthine, varying degrees of disorder were observed across a range of time periods during the formation of hypoxanthine crystals. Using 4D-STEM, we observe a range of similar electron diffraction patterns in xanthine crystals, but this variation is spatial rather than temporal.

Hypoxanthine also has a second form[58], a monoclinic polytype. Although hydrogen-bonding arrangements in both forms are essentially identical, there is greater out-of-plane flexibility in the monoclinic form, resulting in larger canting of the molecules within the layers (Supplementary Note 7, Fig. S7.1). Layers in this polytype are related by a translation vector (Fig. 8ii **C**) with no rotation. The vectors describing the relationship between hypoxanthine layers in this form are different from those found in the triclinic form (Fig. 8ii **A, B**), whereas Form I xanthine is built from a subset of the translations seen in Form II. The variation in stacking possibilities for

both structures is a consequence of a likely shallow energy landscape that is a feature of these layered materials.

Next, we consider the stacking of the hydrogen-bonded layers in guanine. Guanine, one of the purine base pairs, which make up deoxyribonucleic acid (DNA), has two known forms, which are also polytypes of each other[59]. Each form has a stacking sequence that can be described exclusively using translation vectors. Layers in α guanine are related by applying one vector (Fig. 8iii **A**), whilst layers in β guanine are related by applying another vector (Fig. 8iii **B**). Combinations using both **A** and **B** stacking might be possible, but are not observed in currently known crystal forms, suggesting that this combination is a less energetically favourable motif. This may explain why streaking in ED data from guanine is not always observed[2], given that the origin of streaking in xanthine likely comes from the variation of stacking sequence. The dependence of SED on crystal orientation to observe the streaking further supports the additional use of 3D-ED. However, as in hypoxanthine, streaking in biogenic guanine crystals is observed when crystals are developing[35]. Mature crystals exhibited non-streaked, single-crystal diffraction patterns, pointing to a gradual increase in order as molecules of guanine preferentially oriented along one dimension arrange into stacked layers. Whilst this variation is seen across individual

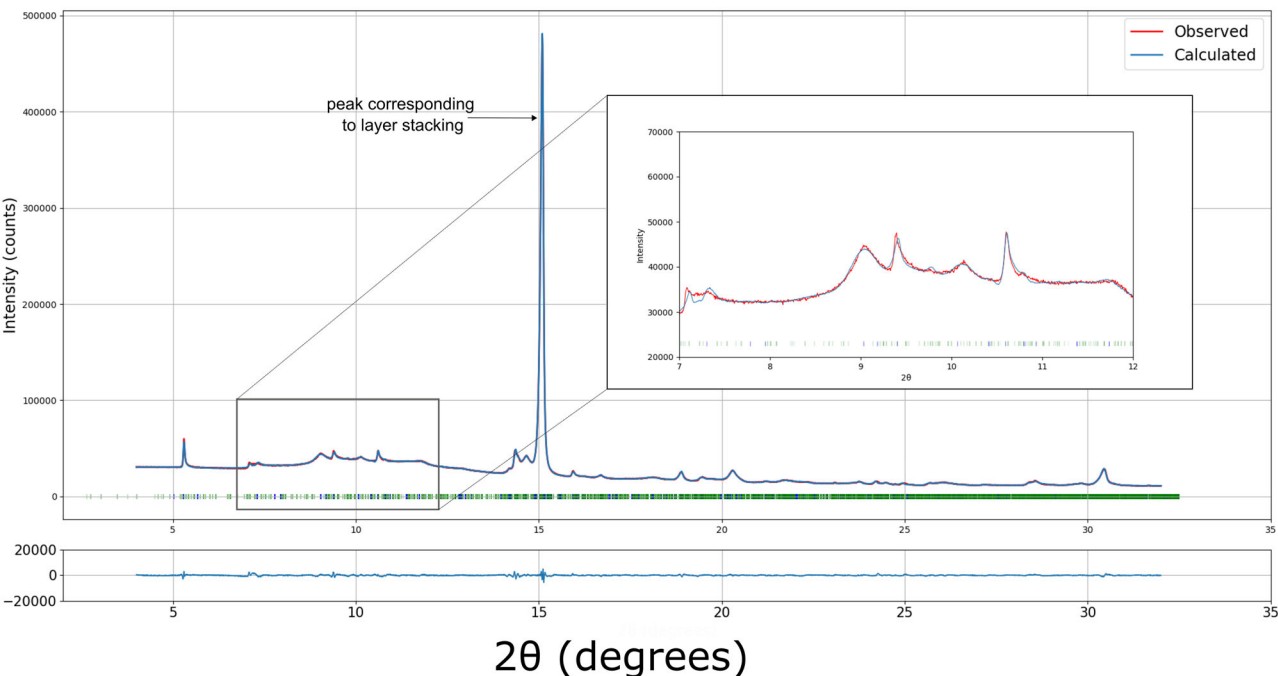

**Fig. 7 | Rietveld refinement on XRPD data from xanthine.** A supercell multiphase Rietveld refinement produced a convincing fit to high-resolution XRPD data, with stacking faults modelled. Expected peak positions for the two phases are shown by the ticks in blue (Form I), green (supercell of Form II). The supercell model leads to a larger number of expected peak positions. The inset shows an improved fit to the asymmetric peak profiles observed compared to a model with perfect lattices, shown in Fig. S6.1. This refines to an $R_{wp}$ of 1.55% and GooF of 2.46%.

**Fig. 8 | Comparing stacking in similar planar crystal structures. i** Stacking sequences in xanthine (where **A**, **B**, **C**, and **D** all appear in Form II) are more varied than those seen in (**ii**). In hypoxanthine, rotation and translations as shown by A and B are present in its triclinic form. However, in monoclinic hypoxanthine, interlayer stacking (vector **C**) is similar to twinning action in xanthine. **iii** guanine polytypes (only **A** appears as the motif in α guanine, whilst only **B** appears as the motif in β guanine). Interlayer vectors are defined by using the shortest magnitude vector.

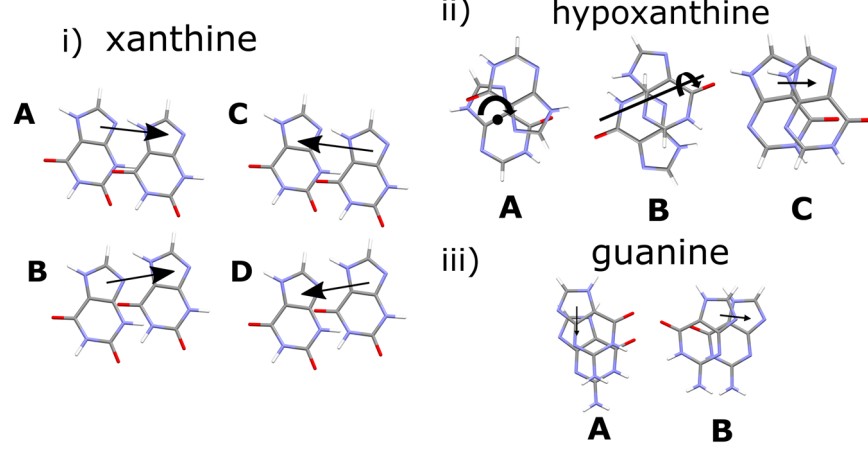

xanthine particles, it is also possible that defects in guanine and hypoxanthine remain 'hidden' from electron diffraction data, as streaking may also not be visible unless the particles of interest are viewed in a direction normal to the layered planes[37]. Finally, domains in biogenic guanine are observed with similar thicknesses and shapes, likely biologically controlled[2]. This is consistent with the regular thickness of xanthine domains observed in Figs. 6 and S5.1. However, domains of xanthine are otherwise less uniform in shape.

Not all molecules with planar crystal structures have polytypes. For example, hydrogen-bonding networks in adenine are different between polymorphic forms, made possible due to the diversity of stable hydrogen-bonding interactions[60]. As another example, both known crystal structures of caffeine are disordered, in which intra-layer molecular positions are not fixed and there are multiple sites of partial occupancy in the unit cell[61,62]. Although these crystal structures are not known to display polytypism, they could still display evidence of planar disorder through disruption of stacking sequences or twinning. These systems are obvious candidates for further investigation using 3D-ED and 4D-STEM. Additional degrees of structural complexity may be included when considering that molecular configurations within layers may also alternate.

Beyond small molecule crystals, polymer crystal systems may also exhibit planar disorder in which streaking has been previously observed[40]. This behaviour often falls under the wider umbrella of mosaicity and may be regarded as a barrier to structure solution. The extended application of similar electron diffraction techniques used in this work may help to understand the nanoscale structure of larger complexes[63], i.e., in cases where the polytypism affects any crystal large enough for single-crystal X-ray diffraction. Finally, further investigation to understand the preferred tendencies of these planar crystal systems could be undertaken using computational Crystal Structure Prediction[64,65] methods such as DFT[33] or Ab Initio

Random Structure Searching[66]. It would also be interesting to understand how this stacking behaviour changes in the hydrated forms of the compounds[67]. This is a topic of extensive interest given the role of nucleic acid interaction in stability and conformational variability of biological systems. However, it is worth highlighting that cryogenic electron diffraction of hydrated structures would require careful sample preparation, as it is possible that structural changes may occur during the freezing process. Within biological systems, most interactions between nucleic acid molecules occur in the presence of solvent molecules, and the successful competition of water may significantly change intermolecular hydrogen bonding[60] and therefore stacking variability[68,69].

## Conclusions

In this work, planar variation in the form of polytypes, twins, and disorder in xanthine were characterised using multi-dimensional electron diffraction modalities. By comparing relative interlayer translation vectors between different polytypes, we reveal how Form I and Form II are closely related. This microstructure is pervasive on the nanoscale to the extent that multiple forms may be present within a single sub-micron particle, as shown using 4D-STEM. With XRPD data, we confirm the co-existence of these two polytypes, and we show that further stacking faults modelling is necessary to fit the data satisfactorily. We believe the methods and findings presented in this work extend beyond xanthine and related purines and apply to a much larger set of organic planar molecules. Via relatively strong supramolecular interactions, many of these small organic molecular crystals form layers held together by weak interactions, which makes them ideal candidates for stacking faults and polytypism. Furthermore, solid solutions doped with different molecules[70] could also exhibit similar behaviour.

In addition to advancing our understanding of planar molecular crystals, this area of study could also have wider implications in the pharmaceutical industry, where there is a regulatory requirement to understand and characterise polymorphic forms[71]. This is a topic of interest, where pure polytypes exist, given the range of properties that different polymorphic crystal forms may have[72–76]. Given that ED has shown a range of crystalline orders is possible within individual particles of xanthine, further thought is needed on the way that the solid-state landscapes of layered compounds, more generally, are described: the interchangeability between different polytypes at the nanoscale introduces an inherent complexity to defining and grouping discrete polymorphs.

In conclusion, we combine the use of 3D-ED and 4D-STEM to highlight the potential for further study of molecular crystal microstructure. More fundamental questions such as possible defect-linked mechanisms of crystal growth could also be answered in future work.

## Methods

### Materials
Xanthine powder was purchased from Sigma Aldrich, X7375, batch WXBD7599V.

### 3D-electron diffraction
Grids were prepared as in previous work, also described in Supplementary Note 1[42]. Continuous rotation 3D-ED was performed using a Thermo Fisher Titan Krios G3i electron microscope operated at 300 kV under cryogenic conditions. A diffraction pattern was recorded for each tilt increment over a range of ±60° at a continuous tilt rate of 1° s$^{-1}$. Diffraction patterns were recorded on a CETA-16M camera with an exposure time of 0.5 s per frame, forming a tilt series of 240 diffraction patterns. These conditions result in a cumulative dose of 20 e Å$^{-2}$. The camera length was set such that Bragg spots corresponding to a resolution up to 0.7 Å could be detected. EPU-D software was used for the acquisition, making use of the auto-eucentric height function to minimise sample movement when tilting to high angles. 3D-ED data were collected from crystals with a cross-section of ca. 1 μm × 1 μm. Data were indexed and integrated using CrysAlisPro 1.171.43.110a (Rigaku Oxford Diffraction, 2024)[77]. The raw electron

diffraction data can be found here: https://zenodo.org/records/15858784[78]. The new experimental crystal structure (of Form II xanthine) described in this manuscript is included in the Supplementary Data File and available in the CCDC with the deposition number:2425886.

### 4D-STEM
4D-STEM microscopy was performed using a Thermo Fisher Spectra 300 microscope operated at 200 kV. Diffraction patterns were acquired using a Medipix3 direct electron detector. A 0.1 mrad convergence angle was used, and the camera length was set such that a resolution of 0.7 Å could be measured. The beam current was 2 pA with a 1 ms dwell time, leading to a cumulative dose per scan of ~10 electrons Å$^{-2}$. Analysis of 4D-STEM data were carried out using hyperspy and pyxem[79] libraries.

### XRPD
In-situ XRPD was performed on Beamline I11 at Diamond Light Source[80,81]. Xanthine powder was loaded into a 0.5 mm borosilicate glass capillary and analysed using an X-ray beam of 0.82408 Å wavelength (15 keV energy), refined using a NIST SRM640c Si standard. The sample was cooled to 80 K, consistent with 3D-ED measurement conditions, using a Cryostream Plus. Measurements were made using the Mythen wide-angle position-sensitive detector.

## Data availability

The data supporting this study are available within the supplementary information files or in public repositories. Specifically, the crystallographic information file for xanthine Form II is available as a supplementary file (Supplementary Data 1) and has also been deposited in the CCDC with the deposition number:2425886. Raw 3D-ED frames used to achieve this structure solution are available on Zenodo (https://zenodo.org/records/15858784)[78]. Finally, the input files used to perform Rietveld refinement with TOPAS are provided in the supplementary files: (1) the model including Form I and Form II xanthine crystal structures (Supplementary Data 2), (2) Form I and Form II xanthine crystal structures with the additional supercell stacking faults model (Supplementary Data 3).

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

## Acknowledgements

H.W.L. and P.A.M. acknowledge funding from the Engineering and Physical Sciences Research Council (Nos. EP/W522120/1 and EP/R008779/1) and a GSK ICASE studentship (grant no. 210193). We also thank Diamond Light Source for access and support in using the I11 beam (award no. CY34800) which contributed to the results presented here. We thank Rigaku for access to the CrysAlisPro software suite.

## Author contributions

H.W.L. prepared the samples, collected the 3D ED and 4D-STEM data, solved and refined the structure with ED data, collected powder X-ray data and performed the Rietveld refinement. R.C.B.C. solved and refined the structure with 3D-ED data. G.I.L. collected powder X-ray data and performed the Rietveld refinement. S.J.D. and L.K.S. were beamline scientists at I11 and assisted with the collection of PXRD data. D.N.J. contributed to discussion at all stages of this research. P.A.M. contributed to discussion and direction and supervised all stages of this research. H.W.L. wrote the manuscript, with feedback and contributions from all authors. All authors have given approval to the final version of the manuscript.

## Competing interests

The authors declare no competing interests.
