## [Transparent Peer Review file · Communications Chemistry]

Polytypes and Planar Defects Revealed in the Purine Base Xanthine using Multi-Dimensional Electron Diffraction

Corresponding Author: Ms Helen Leung

Version 0:

Reviewer comments:

Reviewer #1

(Remarks to the Author)

The manuscript describes the use of electron diffraction modalities to explore the microscopic structure variations in a purine-base, used as an archetype for other layered organic molecular crystals. The experimental sections outline the identification of three key structure variations, twinning, polytypism and stacking faults. The identification of the polytype that seemingly only exists in nanoscale and highly disordered form is significant in the wider analysis of this material. In general the experimental data is very well presented and the results shown strongly support the structure analysis presented. In particular it is pleasing to see how the micro- to nanoscale structure variants observed are used to improve the Rietveld refinement of PXRD data from ensemble samples of meaningfully large numbers of crystals. There are still insufficient examples of the way that different crystallographic methods can be used in a complimentary way, so this is a very important result to include.

The discussion proposes the extension of the approaches used in the article to other similar layered organic crystals given the limited nanoscale information that exists for these materials. There is perhaps, some further discussion on the limitations and challenges this might bring (see later comments). However overall I feel that the manuscript provides a strong support for this proposed work and is a step in the ongoing integration of electron microscopy and electron diffraction in the study of soft and organic materials.

I have a small number of specific queries and comments I would like the authors to consider:

Line 146 (Caption for Figure 3). I found the comment about the change in number of reflections with symmetry a little bit misleading, the additional reflections arise because the orthorhombic unit cell is larger so in the same volume of reciprocal space there are more reflections. If anything increasing symmetry tends to reduce the number of reflections.

Line 149. For clarity do the reciprocal space sections shown in Figure 3 contain reflections associated with both type I and type II structures? It isn't explicit if this is the case from the text, Further to this, is the orthorhombic structure solved against all of the reflections in the patterns, or only from the unique subset from the type II (or at least the 'not-obviously-type-I reflections). A clearer explanation of how the diffraction data was used would be helpful here.

Line 167 Figure 4. Please can you make the axis labels in parts a) and c) more legible. I think also having labelled crystal axes on parts b) and d) would be useful to understand how the structures fit to the H-bonded layers.

Line 212. How the the patterns that are averaged together chosen? Is it just a local binning or is there some guided method (either manually or by an ML approach). Perhaps some indication on one of the Figures about the summed/averaged regions would be helpful for those looking to apply this approach.

Line 340. If the streaking is hard to see in individual patterns, surely this is a strong argument for 3D-ED, I wonder why you haven't emphasised this point?

Line 375. The challenge with hydrated structures is how to prevent the high vacuum and high energy deposition from the beam affecting the structures. Macromolecule studies in TEM typically use cryo freezing and overcome the resulting data limitations through statistics, as all the molecules are approximately identical. I think the variability of individual crystals/stacking sequences would make direct comparison difficult. As a result there is a small but significant difference to the challenges this type of work faces compared to other cryo-EM. I think that perhaps some comment on these challenges to be met would be useful insight for others looking to use these approaches.

Supp Figure 2. The reciprocal space reprojection of the complete data is interpretable for an experienced user of 3D-ED, but I think that reciprocal space sections clearly showing the twinning would be useful to make the results presented in both the main manuscript and the SI more understandable to a wider audience. This would support the twinning diagrams presented in Figure 2

Finally Line 129. Figure 2a, the crystal appears to be imaged at quite a large defocus, I assume that this is in order to achieve meaningful contrast in the TEM image? I appreciate you might not be able to do anything about this but are you confident that the microscope alignment was suitable to record the 3D-ED data without distortion?

Reviewer #2

(Remarks to the Author)

Leung et al. report a crystallographic analysis of polytype formation in a xanthine-based sample using a 4D electron probe. Authors have utilized a nanobeam electron probe to map the mosaicity of crystallites in a commercial xanthine-based sample. The manuscript is well written, and conclusions are consistent with the experimental results. Although the quality of experimental work (4D-STEM) is good, I do not find this work original enough for publication in 'Communications Chemistry'. Additionally, major revisions are needed before publication to a high impact journal. In more details:

- Experimental details regarding the synthesis of the sample studied are missing. Authors mentioned that this is a commercial sample which is fine, but the real question is how do the synthetic conditions affect quality of crystals and therefore conclusions of this study. For example, where the crystals grinded by the manufacturer, is this a slow or fast synthesis? All these conditions might affect the concentration of stacking faults and twin domains.
- Given that the second form of Xanthine shows Bragg reflections in the powder diffraction pattern, can you still claim that this a disordered system as opposed to a distorted or medium ranged ordered crystalline system? I would expect to see a very diffuse scattering signal if this is truly a disordered system. For example, see old study on aspirin at Acta Cryst. (2010). B66, 696–707 (10.1107/S0108768110037055)
- Can the authors fit the strain signal in the powder diffraction data without the use of a supercell? What is the quality of fit?
- The term 'multi-dimensional' in the title is a little misleading since this is a reserved term for crystallographic methods that utilize up to 6 dimensions for aperiodic crystals. I recommend changing this term to '4D-STEM'

Reviewer #3

(Remarks to the Author)

The manuscript reports a new polytype of the xanthine molecule based on a combination of electron diffraction, 4D-STEM, and X-ray powder diffraction data. It presents a high-level analytical approach and applies advanced techniques alongside with a solid crystallographic investigation.

Major Points and Suggestions:

- Data Availability: My primary request is that the original electron diffraction (ED) data be made publicly available via a static link, which should be cited within the manuscript. Transparency and reproducibility are essential for evaluating the presented structural models.
- Evaluation of Supplied ED Data: I have received the raw data from the authors and had the opportunity to inspect the crystal quality. Indeed, an orthorhombic metric appears alongside the main monoclinic phase I. For the monoclinic phase, I observed a β angle of 112° , which is significantly larger than the reported value of 107° . In Figure 3a,b, my measurement of the β angle also gives 111° , not 107° as stated. I would ask the authors to verify this discrepancy. The dataset appears consistent, and I do not understand the origin of this difference. Nevertheless, after extracting the intensities myself, I was able to reproduce the correct solution for phase I.
- Minor Phase II and Space Group Assignment: I did not extract intensities for the minor phase II, relying instead on the data provided in the corresponding CIF file. However, I could not follow the reasoning behind assigning the $P2_12_12_1$ space group. Please indicate the systematic absences (extinctions) that justify this choice. While $P2_12_12_1$ is indeed one of the most common space groups, it is typically observed in chiral molecules. Given that xanthine is achiral, this assignment is rather unusual—though it might be justified as an overall symmetry of a higher-order polytype. Please clarify this point in the manuscript.
- Structure Solution and Reproducibility Issues: I attempted to solve the structure of phase II in SHELXD using the hkl list provided, but the solution failed. Were any special SHELXD settings used that are not described in the manuscript? My statistical indicators were slightly different ($R_{int} = 30.99\%$, completeness = 82.1%), which may be due to differences in the way OLEX2 calculates these metrics. Regardless, the inability to reproduce the structure solution is concerning.
- Structure Validation and Energy Minimization: Assuming the reported structure is correct—and that its reproduction simply requires specific phasing parameters—it would still benefit from further validation. The layer geometry is largely the same as in phase I, and the interlayer interaction upon AB-shift also seems plausible. The key new structural feature in phase II is the layer stacking shift BC or B[-A]. I would recommend that the authors perform energy minimization of the full structure to support their findings. Additionally, constructing idealized models with alternative stacking sequences (e.g., BC, ABC, ABD...) and performing energy minimizations on these could be highly informative.
- Structure description: it would be helpful to measure quantitative lateral shift vectors between adjacent layers, ideally broken down into components along well-defined (molecular) in-plane directions, as in Curtis et al., Solid-state packing of conjugated oligomers: from π -stacks to the herringbone structure. J. Am. Chem. Soc. 2004, 126, 4318–4328; Milita et al., Polymorphism in N,N'-dialkyl-naphthalene diimides. J. Mater. Chem. C. 2020, 8, 3097–3112.
- Powder Diffraction: I would be cautious about drawing any conclusions from the powder diffraction data as presented. The quality of the powder pattern is insufficient to validate the proposed structure, particularly in the presence of multiple

polytypes. Given that the crystal used for ED appeared to be of good quality, why is the powder diffraction measurement so poor? Was the sample inhomogeneous?

A related question: What is the solubility of xanthine? Were any recrystallization attempts made to improve sample quality?

- 4D-STEM Results: The 4D-STEM results are aesthetically pleasing and technically impressive, but their purpose remains unclear. Are they intended to demonstrate the coexistence of different stacking sequences within a single crystal?

- Literature: When discussing stacking fault disorder in layered organic crystals, it may be valuable to reference works by Martin Schmidt (University of Frankfurt) on organic pigments.

Version 1:

Reviewer comments:

Reviewer #1

(Remarks to the Author)

I believe that the resubmitted manuscript has addressed my questions satisfactorily, thank you for your clear and thorough responses.

Reviewer #2

(Remarks to the Author)

Revisions were satisfactory. Thank you

Reviewer #3

(Remarks to the Author)

I am fully satisfied with the current state of the manuscript; however, I noticed a small mismatch in the references—please double-check.

Reviewer #1 (Remarks to the Author):

The manuscript describes the use of electron diffraction modalities to explore the microscopic structure variations in a purine-base, used as an archetype for other layered organic molecular crystals. The experimental sections outline the identification of three key structure variations, twinning, polytypism and stacking faults. The identification of the polytype that seemingly only exists in nanoscale and highly disordered form is significant in the wider analysis of this material. In general the experimental data is very well presented and the results shown strongly support the structure analysis presented. In particular it is pleasing to see how the micro- to nanoscale structure variants observed are used to improve the Rietveld refinement of PXRD data from ensemble samples of meaningfully large numbers of crystals. There are still insufficient examples of the way that different crystallographic methods can be used in a complimentary way, so this is a very important result to include.

The discussion proposes the extension of the approaches used in the article to other similar layered organic crystals given the limited nanoscale information that exists for these materials. There is perhaps, some further discussion on the limitations and challenges this might bring (see later comments). However overall I feel that the manuscript provides a strong support for this proposed work and is a step in the ongoing integration of electron microscopy and electron diffraction in the study of soft and organic materials.

We thank reviewer 1 for their time and their detailed comments on our paper. We reply in red to individual comments below. Where we quote changes made to the paper, we also underline and highlight green the changes.

I have a small number of specific queries and comments I would like the authors to consider:

Line 146 (Caption for Figure 3). I found the comment about the change in number of reflections with symmetry a little bit misleading, the additional reflections arise because the orthorhombic unit cell is larger so in the same volume of reciprocal space there are more reflections. If anything increasing symmetry tends to reduce the number of reflections.

Thank you to the reviewer for pointing this out. We have made this correction to the caption in Figure 3, line 147-148.

'The larger unit cell in Form II results in an increase in the number of reflections, with a limited number of reflections overlapping with those of Form I.'

Line 149. For clarity do the reciprocal space sections shown in Figure 3 contain reflections associated with both type I and type II structures? It isn't explicit if this is the case from the text, Further to this, is the orthorhombic structure solved against all of the reflections in the patterns, or only from the unique subset from the type II (or at least the 'not-obviously-type-I reflections). A clearer explanation of how the diffraction data was used would be helpful here.

We used the multi-crystal function in CrysAlispro when integrating this diffraction data. There are 11 overlapped reflections as reported by the software. In the main body of the text, we have added this information in lines 152-154:

'Integration was carried out using the 'dual crystal' mode in CrysAlisPro, which produced a set of intensities accounting for the overlapped reflections from the Form I phase (only 11 out of 455 reflections corresponding to the new orthorhombic form were reported to overlap).'

In lines 146-151 we have also expanded the Figure 3 caption to try and make this clearer. We now write: *'The h0l section from monoclinic Form I (from a separate crystal) is overlaid onto the orthorhombic Form II hk0 section to compare their similarities. The larger unit cell in Form II results in an increase in the number of reflections, with a limited number of reflections overlapping with those of Form I. The inset in (d) shows the micron-sized bi-crystal of xanthine from which this dataset was obtained. Sections shown in this figure come from the Form II domain of the crystal. Reflections from the Form I domain are largely not seen in these sections, except in the far left of the h0l section. The jagged nature of the left straight facet of the particle hints at its bi-crystalline nature.'*

Line 167 Figure 4. Please can you make the axis labels in parts a) and c) more legible. I think also having labelled crystal axes on parts b) and d) would be useful to understand how the structures fit to the H-bonded layers.

We have made the axes in Figure 4 larger and more legible and included the axes in b) and d).

Line 212. How the the patterns that are averaged together chosen? Is it just a local binning or is there some guided method (either manually or by an ML approach). Perhaps some indication on one of the Figures about the summed/averaged regions would be helpful for those looking to apply this approach.

These regions were chosen using a manual approach, exploring the datasets and observing differences in signal across the 4D dataset. We have added this detail into the main text lines 220-222:

'Firstly, diffraction patterns were summed over small local areas of the xanthine particle, chosen by manual inspection of the datasets. This improved the signal-to-noise from a

given area and highlighted any streaking (e.g. Figure 6c), akin to that seen in 3D-ED data (e.g. Figure 3a).'

These regions of interest are shown in Figure 6b) (via the several colourful squares which indicate from where the summed/averaged regions are taken). This detail is highlighted in the Figure 6 caption. *'Diffraction signal from the areas highlighted in different colours are used to form the summed diffraction patterns shown in c-f).'*

Line 340. If the streaking is hard to see in individual patterns, surely this is a strong argument for 3D-ED, I wonder why you haven't emphasised this point?

Thank you to the reviewer their helpful point of view in helping us emphasise the benefit of using both SED and 3D-ED. To emphasise this, we have added (lines 346-347):

'The dependence of SED on crystal orientation to observe the streaking further supports the additional use of 3D-ED.'

Line 375. The challenge with hydrated structures is how to prevent the high vacuum and high energy deposition from the beam affecting the structures. Macromolecule studies in TEM typically use cryo freezing and overcome the resulting data limitations through statistics, as all the molecules are approximately identical. I think the variability of individual crystals/stacking sequences would make direct comparison difficult. As a result there is a small but significant difference to the challenges this type of work faces compared to other cryo-EM. I think that perhaps some comment on these challenges to be met would be useful insight for others looking to use these approaches.

Thank you for the reviewer's insight here. Although our 3D-ED work was also carried out at cryogenic temperature, we have added more context to our comments based on the points of view presented here to reflect the enhanced challenges of working under cryogenic conditions with hydrated samples. We have added in this sentence (lines 386-388):

'However, it is worth highlighting that cryogenic electron diffraction of hydrated structures would require careful sample preparation as it is possible that structural changes may occur during the freezing process.'

Supp Figure 2. The reciprocal space reprojection of the complete data is interpretable for an experienced user of 3D-ED, but I think that reciprocal space sections clearly showing the twinning would be useful to make the results presented in both the main manuscript and the SI more understandable to a wider audience. This would support the twinning diagrams presented in Figure 2

The schematic in Figure 2 shows the preservation of the hydrogen-bonded planes which is crucial to our proposed twinning mechanism. However, we appreciate the referee's suggestion that any further interested readers might like to see the dataset in further detail. As such, we have added a figure into the supplementary information (Figure S3.1) to explain the twinning rotation axis that we observed in more detail, alongside a schematic. We thank the referee for asking us to include this in more detail which has helped us refine our explanation of what we observe: we have now also added an interesting observation which is that any twin rotation axis that lies within the (10-1) hydrogen-bonded planes could in principle lead to alternative twinning which would still act to preserve the hydrogen-bonded planes in xanthine. Figure S3.1 supports this. We add the following supporting text to our figure in lines 58-61 in the supplementary information:

'We note that any 2-fold twin rotation axis (with the direction $[1v1]$, where v can take any value such that the rotation axis lies within the $(10\bar{1})$ hydrogen-bonded planes) could in principle lead to alternative twinning which would still act to preserve the hydrogen-bonded planes in xanthine.'

Finally Line 129. Figure 2a, the crystal appears to be imaged at quite a large defocus, I assume that this is in order to achieve meaningful contrast in the TEM image? I appreciate you might not be able to do anything about this but are you confident that the microscope alignment was suitable to record the 3D-ED data without distortion?

Yes, the defocus was set to large values in this specific imaging mode to maximise contrast given the low dose conditions. However, diffraction alignment was then separately loaded: this alignment setting had been carefully set up and calibrated so we are confident that the diffraction data itself was taken under optimised acquisition and alignment conditions.

Reviewer #2 (Remarks to the Author):

Leung et al. report a crystallographic analysis of polytype formation in a xanthine-based sample using a 4D electron probe. Authors have utilized a nanobeam electron probe to map the mosaicity of crystallites in a commercial xanthine-bases sample.

The manuscript is well written, and conclusions are consistent with the experimental results. Although the quality of experimental work (4D-STEM) is good, I do not find this work original enough for publication in 'Communications Chemistry'. Additionally, major revisions are needed before publication to a high impact journal. In more details:

We thank Reviewer 2 for their time and detailed comments on our paper. We reply in red to individual comments below. Where we quote changes made to the paper, we also underline and highlight green the changes.

- Experimental details regarding the synthesis of the sample studied are missing. Authors mentioned that this is a commercial sample which is fine, but the real question is how do the synthetic conditions affect quality of crystals and therefore conclusions of this study. For example, where the crystals grinded by the manufacturer, is this a slow or fast synthesis? All these conditions might affect the concertation of stacking faults and twin domains.

We agree with reviewer 2 that this information would give more insight. We have retrieved all the information provided from the commercial manufacture of this sample: in our text we had cited our previous work which contains this information but have now included this explicitly in this paper as well in both the main text and S1.

In the main text lines 107-108: 'Samples from commercially acquired xanthine, synthetically produced by deamination of guanine [Gulevskaya], were deposited onto TEM grids (see S1).'

In the supplementary S1 we have added: 'Attempts to grow sufficiently large single crystals from solution were unsuccessful and resulted in polycrystalline spherulitic aggregates. An aqueous suspension of xanthine powder was made using 0.5 mg of xanthine in 14.5 ml of distilled water. 5 μ L of this suspension was micro-pipetted and dropped directly onto Quantifoil grids. Grids were left in a fume cupboard to allow the water to evaporate under room temperature conditions. This left behind more evenly distributed xanthine crystals sufficiently isolated for 3D-ED.'

However, we do not know if xanthine particles were ground by the manufacturer, or more specific synthetic conditions than what is already provided. We state in our sample prep that we do not ourselves grind the crystals (previously in this was in reference to our other work but we have also included this detail directly as above). As a result, we do

not suggest a link in this paper between the manufacturing conditions and the microstructure we observe in these synthetic xanthine samples.

- Given that the second form of Xanthine shows Bragg reflections in the powder diffraction pattern, can you still claim that this a disordered system as opposed to a distorted or medium ranged ordered crystalline system? I would expect to see a very diffuse scattering signal if this is truly a disordered system. For example, see old study on aspirin at Acta Cryst. (2010). B66, 696–707 (10.1107/S0108768110037055)

In this work, we refer to microstructural planar disorder, not atomic disorder. Such a (planar) disordered system would be expected to show Bragg peaks (Dornberger-Schiff, 1956, doi: 10.1107/S0365110X56001625). The peak asymmetry in the XRPD clearly points to disorder and in the paper, we show that we can model this with stacking faults based on insights we gain about the structures from 3D-ED.

- Can the authors fit the strain signal in the powder diffraction data without the use of a supercell? What is the quality of fit?

We performed a Rietveld analysis without the use of a supercell in the SI (Figure S6.1). As reported in both the main and supplementary text, this gave an Rwp of 2.36%, and Goodness of Fit (GooF) of 3.72%, with all peak maxima accounted for (Figure S6.1). However, this more conventional structural model does not model the peak profile as accurately as our final approach, incorporating the use of a supercell model.

- The term 'multi-dimensional' in the title is a little misleading since this is a reserved term for crystallographic methods that utilize up to 6 dimensions for aperiodic crystals. I recommend changing this term to '4D-STEM'

We use multi-dimensional here in the context of using both 4D-STEM and 3D-ED. In the field of electron microscopy, it is regularly used to describe many modalities. We also note that the term multi-dimensional is used explicitly in reference to diffraction, not to crystals. This term is accompanied by an explanation of the combination of these electron diffraction modalities in the abstract, introduction, and conclusions of this work. Just using '4D-STEM' would miss out the key concept that we have also used 3D-ED.

Reviewer #3 (Remarks to the Author):

The manuscript reports a new polytype of the xanthine molecule based on a combination of electron diffraction, 4D-STEM, and X-ray powder diffraction data. It presents a high-level analytical approach and applies advanced techniques alongside with a solid crystallographic investigation.

We thank Reviewer 3 for taking the time to review our work so thoroughly. We reply in red to further individual comments below. Where we quote changes made to the paper, we also underline and highlight green the changes.

Major Points and Suggestions:

- Data Availability: My primary request is that the original electron diffraction (ED) data be made publicly available via a static link, which should be cited within the manuscript. Transparency and reproducibility are essential for evaluating the presented structural models.

We are very happy to publish our data and have now uploaded it to a Zenodo link which is also now provided within our manuscript when describing the electron diffraction methods (line 431).

- Evaluation of Supplied ED Data: I have received the raw data from the authors and had the opportunity to inspect the crystal quality. Indeed, an orthorhombic metric appears alongside the main monoclinic phase I. For the monoclinic phase, I observed a β angle of 112° , which is significantly larger than the reported value of 107° . In Figure 3a,b, my measurement of the β angle also gives 111° , not 107° as stated. I would ask the authors to verify this discrepancy. The dataset appears consistent, and I do not understand the origin of this difference. Nevertheless, after extracting the intensities myself, I was able to reproduce the correct solution for phase I.

We thank Reviewer 3 for taking the time to review our work so thoroughly. This is to do with the definition of the unit cell so depends simply which angle you choose to measure in the reciprocal space sections. It is crystallographic convention to define a unit cell using angles closest to 90 degrees, hence why our Form I unit cell is defined with 107° rather than 112° . The diagram below explains the referee's observations.

- Minor Phase II and Space Group Assignment: I did not extract intensities for the minor phase II, relying instead on the data provided in the corresponding CIF file. However, I could not follow the reasoning behind assigning the $P2_12_12_1$ space group. Please indicate the systematic absences (extinctions) that justify this choice. While $P2_12_12_1$ is indeed one of the most common space groups, it is typically observed in chiral molecules. Given that xanthine is achiral, this assignment is rather unusual—though it might be justified as an overall symmetry of a higher-order polytype. Please clarify this point in the manuscript.

Enantio-pure chiral molecules must be in a Sohncke space group (which includes $P2_12_12_1$). However, it is possible for non-chiral molecules to occupy the $P2_12_12_1$ space group. We refer to literature, for example: Pidcock, 2005 'Achiral molecules in non-centrosymmetric space groups (doi: 10.1039/B505236J)': in which the author writes: '*Currently it appears that rigid molecules, particularly if they have mirror symmetry, have an increased chance of crystallisation in a Sohncke space group.*', which draws some (limited) parallels with our rigid xanthine molecule.

- Structure Solution and Reproducibility Issues: I attempted to solve the structure of phase II in SHELXD using the hkl list provided, but the solution failed. Were any special SHELXD settings used that are not described in the manuscript? My statistical indicators were slightly different ($R_{int} = 30.99\%$, completeness = 82.1%), which may be due to differences in the way OLEX2 calculates these metrics. Regardless, the inability to reproduce the structure solution is concerning.

Thank you very much to the reviewer for checking this. In order to provide more detail, we referred back to our lab notes. For the final solution, we used SHELXT, not SHELXD. This is incorrectly reported in our main text but correctly reported in the CIF. We have made this correction to the main text. Both SHELXD and SHELXT are dual space methods, so there is no need to change the rest of this sentence in the main text. In addition, we have added in the detailed settings used to arrive to this solution which were '-m500 -s"P2(1)2(1)2(1)'"'. This effectively iterates the initial number of P1 dual space iterations 500 times, and searches for solutions within the $P2_12_12_1$ space group. We have made this correction and added the further detail to the main text (lines 154-157):

'A structure solution for the orthorhombic phase was successfully achieved using ab initio Dual Space methods implemented in SHELXT (with 500 trials, specifying the $P2_12_12_1$ space group), $Z'=4$, $Z=16$, with a 79% completeness.'

- Structure Validation and Energy Minimization: Assuming the reported structure is

correct—and that its reproduction simply requires specific phasing parameters—it would still benefit from further validation. The layer geometry is largely the same as in phase I, and the interlayer interaction upon AB-shift also seems plausible. The key new structural feature in phase II is the layer stacking shift BC or B[−A]. I would recommend that the authors perform energy minimization of the full structure to support their findings. Additionally, constructing idealized models with alternative stacking sequences (e.g., BC, ABC, ABD...) and performing energy minimizations on these could be highly informative.

We agree with the reviewer that DFT would support the validation of this work and this is mentioned in the paper *'Finally, further investigation to understand the preferred tendencies of these planar crystal systems could be undertaken using computational Crystal Structure Prediction^{60,61} methods such as DFT [Ifliand et al.] or Ab Initio Random Structure Searching⁶².'*

To further support this discussion, we have included (in line 382) a new reference to a paper by Ifliand et al. in which the authors carry out DFT in xanthine, (10.26434/chemrxiv-2025-6lnth), taking observations from our previous work on Form I xanthine. We have also added specific reference to this work when we mention shallow energy landscapes in the xanthine structure (lines 171-173):

'Different polytypes may arise when multiple stacking arrangements lie within a shallow energy landscape and there are multiple energetically-favourable configurations³⁰: in xanthine the case of a shallow energy landscape is supported by Density Functional Theory (DFT) modelling approaches for variations on its layered structure⁴.'

We believe the results presented in the paper, alongside the above cited work, is sufficient evidence to support the structures. Further investigation using DFT, whilst useful and insightful, falls beyond the scope of this work.

- Structure description: it would be helpful to measure quantitative lateral shift vectors between adjacent layers, ideally broken down into components along well-defined (molecular) in-plane directions, as in Curtis et al., Solid-state packing of conjugated oligomers: from π -stacks to the herringbone structure. J. Am. Chem. Soc. 2004, 126, 4318–4328; Milita et al., Polymorphism in N,N'-dialkyl-naphthalene diimides. J. Mater. Chem. C. 2020, 8, 3097–3112.

We describe the shifts in the caption of Figure 5 in the main text, describing the magnitudes of **a** and **c** components (as quantitative lateral shift vectors- (0.377**a**, 0.250**b**, 0.041**c**)).

- Powder Diffraction: I would be cautious about drawing any conclusions from the powder diffraction data as presented. The quality of the powder pattern is insufficient to

validate the proposed structure, particularly in the presence of multiple polytypes. Given that the crystal used for ED appeared to be of good quality, why is the powder diffraction measurement so poor? Was the sample inhomogeneous?

The XRPD data is from the synchrotron, so this is already high-resolution data. Broadness in the pattern is therefore a feature of the sample, with the asymmetry of the peaks explained by the stacking model. Peaks associated with the planar stacking (such as the strongest peak) are not asymmetrical which supports that only some peaks are affected by asymmetric broadening- again another feature of the powder. The powder is likely composed of multiple forms of xanthine: we demonstrate the possibilities of different crystal forms across multiple crystals of xanthine in our SED data which explains why the powder is inhomogeneous compared to the crystals used for ED structure solution. We show the powder fit (with three different models: 1) with only Form I, 2) with both Form I and II, 3) with Form I, Form II, and a stacking fault model). The improved R-factors with the latter model support our work.

A related question: What is the solubility of xanthine? Were any recrystallization attempts made to improve sample quality?

Attempts at re-crystallisation of xanthine resulted in large polycrystalline particles (this has been added to S1 as requested by referee 1 above). We found that creating an aqueous dispersion (below the solubility limit of xanthine) helped evenly disperse the xanthine crystals on the grid for our experiments. We previously included a reference to the sample preparation methods in our previous work, but we have now explicitly included this description in this paper for easier reference for the reader in S1 (supplementary information, lines 20-26).

In the supplementary information, S1, we have added: *'Attempts to grow sufficiently large single crystals from solution were unsuccessful and resulted in polycrystalline spherulitic aggregates. An aqueous suspension of xanthine powder was made using 0.5 mg of xanthine in 14.5 ml of distilled water. 5 μ L of this suspension was micro-pipetted and dropped directly onto Quantifoil grids. Grids were left in a fume cupboard to allow the water to evaporate under room temperature conditions. This left behind more evenly distributed xanthine crystals sufficiently isolated for 3D-ED.'*

- 4D-STEM Results: The 4D-STEM results are aesthetically pleasing and technically impressive, but their purpose remains unclear. Are they intended to demonstrate the coexistence of different stacking sequences within a single crystal?

The purpose of 4D-STEM data is to show the heterogenous nature of the xanthine structures (especially the planar defects), even within individual crystals which cause the wider 'disorder' seen in the XRPD data via asymmetrical peak shapes, despite good individual 3D-ED structure solutions. The 3D-ED crystal structures are used to explain the possible magnitude and direction of the stacking fault disorder observed in the 4D-STEM data. In the discussion section, we highlight that we can see a range of order within individual xanthine crystals- something that cannot be done with 3D-ED alone. However, we have prefaced the sentence with 'Using 4D-STEM' to explicitly state this can be seen from application of this technique.

Line 325: ‘.. Using 4D-STEM, we observe a range of similar electron diffraction patterns in xanthine crystals, but this variation is spatial rather than temporal.’

- Literature: When discussing stacking fault disorder in layered organic crystals, it may be valuable to reference works by Martin Schmidt (University of Frankfurt) on organic pigments.

We thank the referee for their insight here and have included reference to some of this work where stacking disorder is discussed as a potential reason experimental and simulated XRPD profiles showing differences (10.1021/jacs.3c14800). We also include reference to Gorelik et al. where layers exhibit stacking disorder with a mixture of herringbone and parallel arrangements which have parallels with our work, again in organic pigments.

Line 69-70: ‘Stacking disorder has also previously been suggested as the reason for differences between experimental and simulated XRPD profiles in layered organic pigments [Gorelik et al., Krysiak et al.]’

Line 222-224: ‘The streaking observed is consistent with that previously seen in guanine³², layered organic pigments [Gorelik et al., Krysiak et al.]’, and other systems³³⁻³⁷.